# A Study of the Spatial–Temporal Development Patterns and Influencing Factors of China's National Archaeological Site Parks

**Yueting Xi [1], Taili Liu [1], Siliang Chen [2],\*⬤, Xinru Zhang [1], Suyi Qu [1] and Yue Dong [2]**

1 School of Economics and Management, Chang'an University, Middle Section, Nan'er Huan Road, Xi'an 710064, China; yuetingxi@chd.edu.cn (Y.X.); liutaili@chd.edu.com (T.L.); zhangxinru@chd.edu.cn (X.Z.); qusuyi@chd.edu.cn (S.Q.)

2 School of Architecture, Chang'an University, No. 161, Chang'an Road, Xi'an 710061, China; yuedong@chd.edu.cn

\* Correspondence: chensiliang@chd.edu.cn; Tel.: +86-133-2450-9206

**Abstract:** National Archaeological Site Parks are strategic projects in China for innovatively exploring the scientific protection and rational utilization of large heritage sites for the purpose of balancing urban development and protecting cultural heritage. Therefore, research on the spatial–temporal evolution and influencing factors of National Archaeological Site Parks can improve and optimize their management and pattern systems and is therefore of great significance for the sustainable development of large sites and their surrounding areas. Targeting the 135 National Archaeological Site Parks of China, this paper adopts the nearest-neighbor index analytical method, the kernel density estimation method, the standard deviation ellipse method, the method of constructing an indicator system, and an SPSS factor analysis method to analyze the spatial–temporal evolution and influencing factors of these parks. The findings are as follows: (1) In terms of the temporal evolution, the number of National Archaeological Site Parks increased from 2010 to 2022, and the ages and types of the large heritage sites they are built upon gradually became more balanced and diverse. (2) In terms of the spatial evolution, National Archaeological Site Parks form two high-density areas in Central China and East China. Their overall distribution is consistent with China's "Hu Line". (3) In terms of spatial–temporal evolution, the center of gravity of National Archaeological Site Parks' distribution is advancing toward Southwest China, and the trend of a more balanced distribution is rising. (4) Finally, regional development, heritage tourism, historical resources, and government support were observed to be factors that affect the spatial–temporal patterns of the National Archaeological Site Parks. Based on these findings, we propose specific strategies for coordinating and linking the above four major influencing factors to promote the rational utilization of large heritage sites and the sustainable development of National Archaeological Site Parks. We aim to improve and optimize the management and pattern systems of National Archaeological Site Parks, to promote urban renewal based on large heritage sites, and to provide valuable insights for policy makers and heritage practitioners in other countries with historical resources around the world.

**Keywords:** China; protection of large heritage sites; National Archaeological Site Park; systems for managing cultural heritage; sustainable development; spatial–temporal evolution; influencing factors

## 1. Introduction

The United Nations 2030 Agenda for Sustainable Development recognizes the role of culture and cultural heritage in sustainability and proposes "strengthening efforts to protect and safeguard the world's cultural and natural heritage" [1]. In 2021, an initiative of the Sustainable Development Goals Working Group of the International Council on Monuments and Sites identified heritage as the basis for achieving the United Nations' Sustainable Development Goals (SDGs) [2]. This document demonstrates that the protection and utilization of cultural heritage have a positive impact on sustainable social, cultural, and

economic development. One of the important categories of cultural heritage is heritage sites, which are defined by the UNESCO Convention Concerning the Protection of the World Cultural and Natural Heritage as human works or joint works of nature and humans and archaeological sites of outstanding universal value from a historical, aesthetic, ethnographic, or anthropological point of view [3]. A large number of publications have reported on the conservation and utilization practices of ancient sites in many regions as well as the associated difficulties [4–8]. In countries around the world, ancient sites play important roles, for example as tourist attractions and social assets [9], carriers of culture and memory, and contributors to the revitalization of historic cities [10], the development of tourism [11], and the education of the public [12]. The potential value of these sites is also realized through their adaptive reuse [13,14], contributing to sustainable local development.

In China, large heritage sites refer to large-scale ancient cultural sites with outstanding cultural value in Chinese heritage; these sites often contain cultural relics and reflect regional attributes of the natural and social characteristics of the area in which they are located [15]. The National Archaeological Site Parks are sites facilitating China's active exploration of the scientific protection and rational utilization of large-scale archaeological sites under the pressure of rapid economic development and social transformation; they were implemented as part of the official Chinese cultural heritage management system of large heritage sites in the National Archaeological Site Park Management Measures [16] document in 2009. This effort has greatly improved the once passive and lagging process of the extensive protection and management of large historical sites in China [17]. At the same time, they have also responded to new international standards for the treatment of archaeological sites; improved the material conditions of the surrounding environment in terms of economics, space, and facilities; and promoted the transformation of the functions of large heritage sites from focusing solely on "cultural inheritance" to benefiting "regional development" [18]. These changes have increased the public awareness of the fact that culture can promote and facilitate economic development, especially in developing countries [19]. The years of exploration and implementation of China's National Archaeological Site Parks have provided an approach with practical significance and operational value to large heritage sites' protection and management. In 2021, the Outline of the Fourteenth Five-Year Plan for the National Economic and Social Development of the People's Republic of China and Vision 2035 [20] emphasized the "strengthening of the protection and use of cultural relics" and called for the "promotion of the innovative development of cultural heritage tourism". For the first time, archaeological site parks were included in the report, which suggests that archaeological site parks have become a strategic initiative for balancing sustainable urban development and heritage conservation in all regions of China.

As an important system for cultural heritage management and a tool for sustainable urban development, archaeological site parks have received extensive attention from scholars and policy makers. Mainly focusing on the theoretical construction of archaeological site parks [21], scientific planning [22], archaeological work [23], and environmental remediation [24], qualitative research conducted by Chinese scholars has laid the foundations for the research and construction of the National Archaeological Site Parks. The Salalah Guidelines [25], promulgated by the International Council on Monuments and Sites in 2015, recommends the inclusion of "archaeological site parks" in the official terminology of the international cultural heritage field, and their implementation as a tool capable of organically linking site protection efforts, scientific research, and the public. Most scholars tend to choose National Archaeological Site Parks comprising representative and well-developed areas to conduct research on the display methods [26], management and operation [27], evaluation systems [28], heritage tourism development [29], spatial development models [30], etc., of these sites. Significant research results have been obtained regarding the protection and utilization of heritage in the context of archaeological site parks. The Salalah Guidelines for the Management of Public Archaeological Sites were officially adopted at the 19th General Assembly of the International Council on Monuments and Sites in 2017. These important guidelines provide specific guidance for the sustainable management of

archaeological sites [31], and also promote the development of archaeological site parks in the international community. The international community's research on archaeological site parks has shifted from emphasizing material protection to embodying the human dimension in management practices [32] and has gradually started to focus on the broader local social value of archaeological site parks [33] and their contributions to the Sustainable Development Goals [34]. Some common findings of previous studies are of reference value for this research. These findings share the view that the construction of China's National Archaeological Site Parks involves various issues such as cultural heritage preservation, urban and rural construction, tourism development, and so on. Attention has been paid to the relationship between archaeological site parks, urban renewal, and the regional society. Despite China's long history of curating relics, the implementation of an institutional system responsible for the protection and exhibition of archaeological heritage is a relatively recent phenomenon in the Chinese cultural landscape. Previous studies have primarily analyzed National Archaeological Site Parks from the perspectives of archaeology, history, and management [26,28,30], mostly focusing on specific regions or specific archaeological site parks and adopting a single analytical method [26,33,34]. Research providing a systematic observation and macroscopic understanding of National Archaeological Site Parks from a geographical perspective is rare. This lack of systematic and complete studies of the spatial–temporal evolution, distributional management system characteristics, and influencing mechanisms of the National Archaeological Site Park management system on a large scale is a weakness of the existing literature.

Based on the above text, this paper targets 135 National Archaeological Site Parks in China, comprehensively and deeply studying their spatial–temporal patterns and influencing factors from a geographical perspective and on the "national" scale. Given that China's large heritage sites are characterized by their long history, wide distribution, large number, and complex types, and that their cumulative temporal and spatial local attributes are impacted by the imbalanced development of the regions in which they are located, which also experience continuous changes in this dynamic process, the spatial–temporal evolution and influencing factors of China's National Archaeological Site Parks are complex. This study adopts methods of GIS analysis, indicator system construction, and SPSS factor analysis to explore the spatial–temporal evolution patterns and influencing mechanisms of China's National Archaeological Parks formally proposed so far (2010–2022). We identify three main research objectives: (1) analyzing the spatial–temporal evolution patterns of National Archaeological Site Parks by using GIS spatial analysis tools and visualization functions; (2) identifying the main driving factors of the spatial–temporal distribution of China's National Archaeological Site Parks by constructing an indicator system and using statistical analysis methods; and (3) proposing specific strategies for coordinating and linking the four major influencing factors to promote the rational utilization of large heritage sites and the sustainable development of National Archaeological Site Parks.

The aim of this study is to reveal the spatial–temporal evolution patterns and influencing factors of China's National Archaeological Site Parks from 2010 to 2022 on a national scale, so as to improve the management and pattern systems of China's National Archaeological Site Parks and provide a scientific basis for their sustainable development under the context of urban renewal. This paper presents a case study of China's National Archaeological Site Parks, aiming to provide a useful reference for heritage conservation practices, particularly the conservation of large heritage sites, in other countries.

## 2. Materials and Methods

### 2.1. Study Area and Objects

The National Archaeological Site Parks selected for this study are located in China (73°33′ E to 135°05′, 3°51′ to 53°33′). China's large heritage sites usually overlap with modern cities, and the contradiction between heritage protection and urban development is very prominent. Approximately one-eighth of the total number of large heritage sites (1194) are listed as nationally protected key cultural relics. The State Administration of Cultural

Heritage of China conducted four rounds of assessing and establishing National Archaeological Site Park projects in 2010, 2013, 2017, and 2022. By the end of 2022, 55 National Archaeological Site Parks had been built in China, and 80 projects had been included in the approved project list. These 135 parks are distributed across 27 provinces (in cities and districts), gradually forming a National Archaeological Site Park management system with a wide distribution, a long time span, comprehensive coverage types, and various local conditions (Figure 1). Among them, the 55 projects completed are archaeological site parks that meet the criteria in the Rules for the Assessment of National Archaeological Site Parks, and the other 80 are archaeological site parks approved by the State Administration of Cultural Heritage of China that have begun to take shape. These sites also take into account the safety of large heritage sites and the growing public cultural needs of the people, and represent an important opportunity for China to do a good job of protecting and utilizing large heritage sites and promoting the sustainable development of National Archaeological Site Parks in the future.

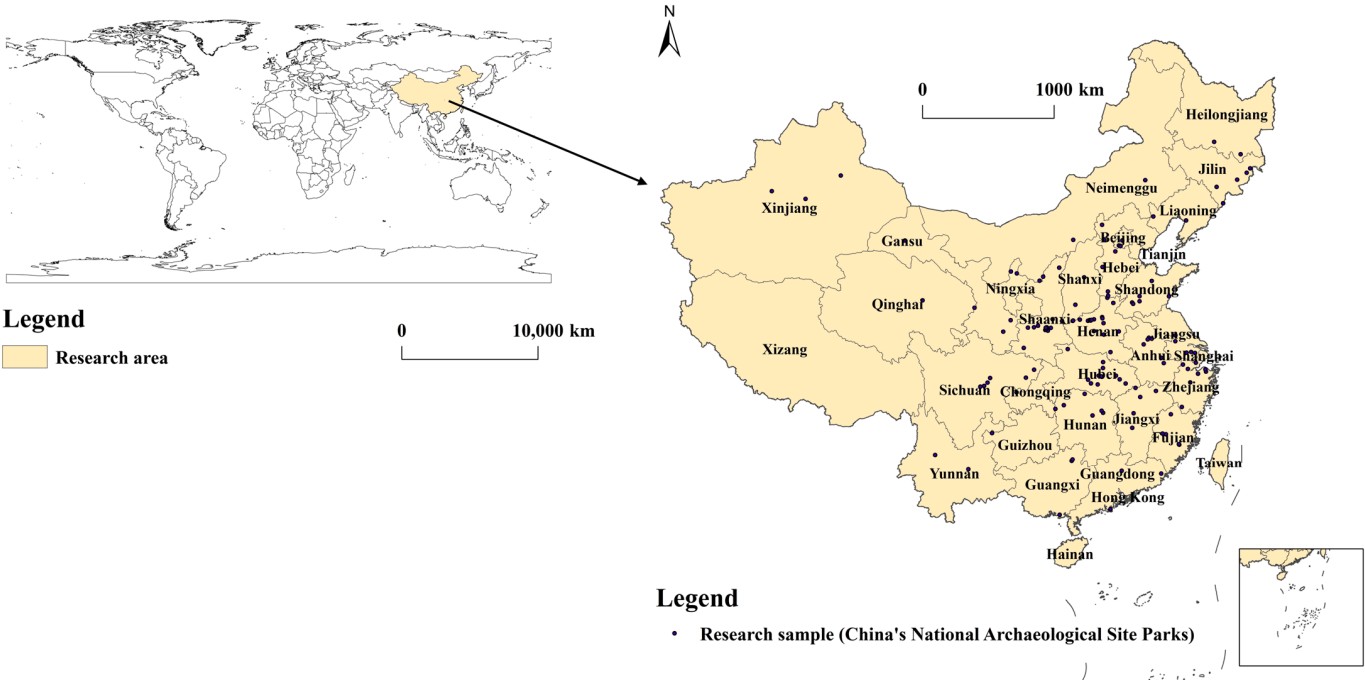

**Figure 1.** Map of the research area and research subjects.

It is pointed out in China's 2022 National Archaeological Site Park Management Measures [35] that the National Archaeological Site Parks focus on important archaeological sites and their surrounding environments. As specific public cultural spaces with national significance related to research and interpretation, the protection and utilization of archaeological sites, and cultural inheritance, National Archaeological Site Parks have the functions of contributing to scientific research, education, and recreation (Figure 2). National Archaeological Site Parks are sorted into eight types [36], namely garden sites, handicraft sites, cave sites, mausoleum sites, urban sites, settlement sites, architectural complex sites, and project sites.

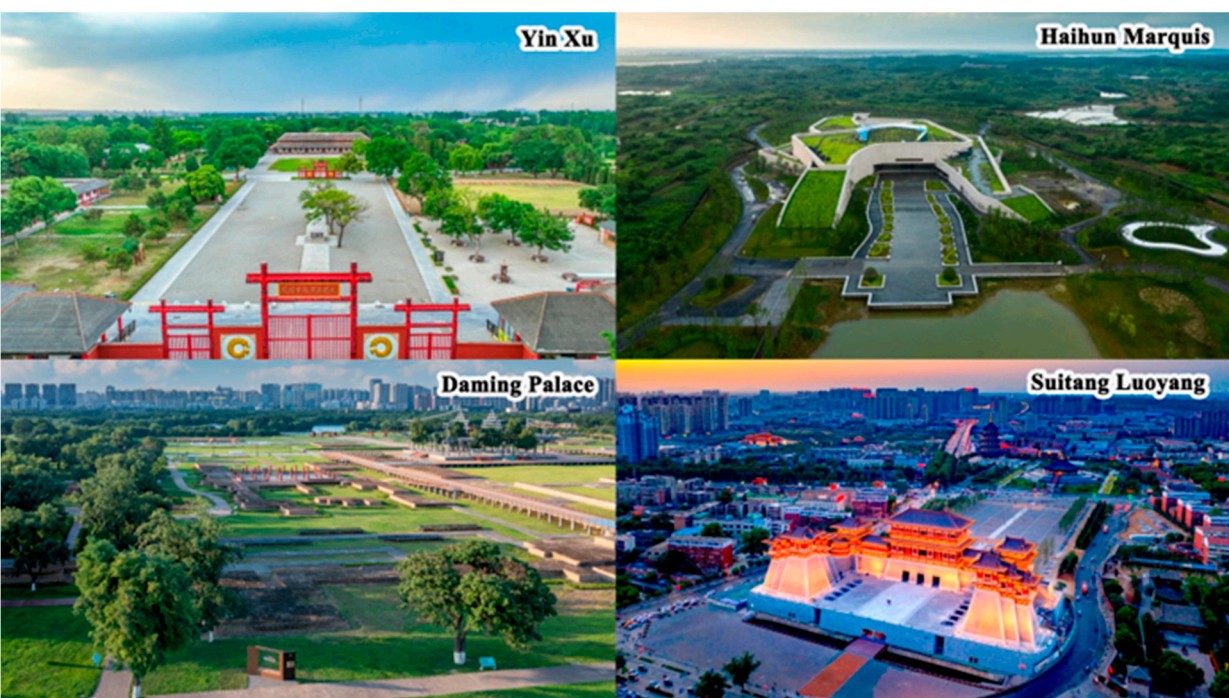

**Figure 2.** China's National Archaeological Site Parks. Please check Supplementary Materials for more information.

### 2.2. Data Sources and Processing

The data sources and processing methods used in this study mainly include the following aspects: (1) The National Cultural Heritage State Administration of China website (http://www.ncha.gov.cn/index.html, accessed on 6 January 2024) was used to query the assessment list of China's National Archaeological Site Parks. Referring to relevant information such as the *Atlas of Chinese Cultural Relics of China's National Cultural Heritage State Administration* and consulting archaeological site protection experts, the type, age, and geographical location of and other information on each National Archaeological Site Park was obtained. (2) The geographic coordinates of 135 National Archaeological Site Parks were collected with the help of Baidu Maps and AMAP according to geographical location information. The data processing function of ArcGIS10.8 software was used to project and check the coordinate data. After processing, the data were exported into the WGS1984 coordinate system, summarized, and then input into the GIS system to establish a spatial point database. Among the spatial data, a map of China and a world map were downloaded from the website of the Ministry of Natural Resources of China (https://www.mnr.gov.cn/, accessed on 6 January 2024), and the maps' approval number is GS (2020) 4619. (3) Finally, data affecting the spatial–temporal patterns of the National Archaeological Site Parks were collected from the 2022 *China Statistical Yearbook* (https://www.stats.gov.cn/, accessed on 6 January 2024). Among these data, the data on the average tourism revenue from 2010 to 2019 were calculated using the entropy method.

### 2.3. Research Methods

This paper adopts GIS analysis, indicator system construction, and SPSS factor analysis as the key methods. After collecting and processing the data, we analyzed the temporal, spatial, and spatial–temporal evolution characteristics of China's National Archaeological Site Parks from 2010 to 2022 and visualized their patterns. Then, we integrated national-scale resource and environmental indicators and social development indicators, among others, to identify the influencing factors (Figure 3).

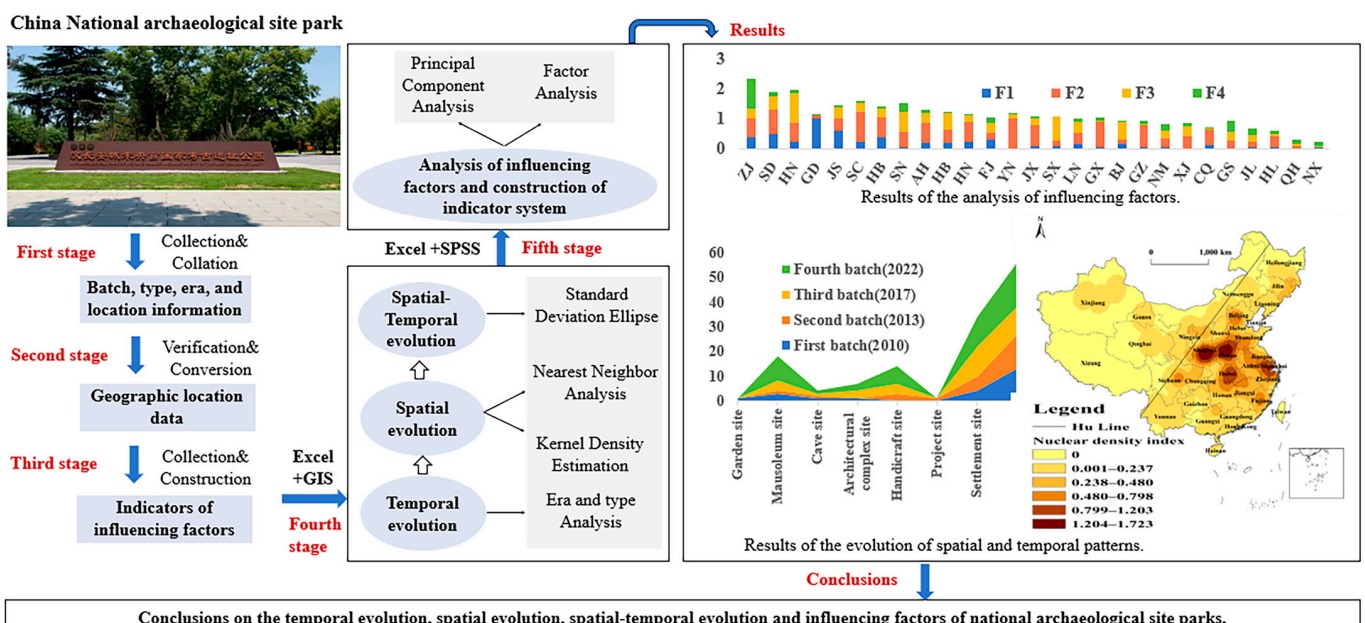

**Figure 3.** Research methodology and pathway flowchart.

2.3.1. Average Nearest-Neighbor Method

The nearest-neighbor method is used to measure the distribution patterns of "point elements" in geospatial space, determining their geospatial structural patterns through the degree of mutual proximity between research objects. This paper analyzes the proximity of National Archaeological Site Parks in China and identifies their spatial distribution patterns and types. The nearest-neighbor indicator can be calculated using the following formula [37]:

$$\overline{r_E} = \frac{1}{2\sqrt{\frac{n}{A}}}, R = \frac{\overline{r_1}}{\overline{r_E}}$$

where R is the nearest-neighbor indicator; $r_1$ is the actual nearest-neighbor distance in space; $r_E$ is the theoretical nearest-neighbor distance; A is the research area; and n is the number of research objects in the area.

2.3.2. Kernel Density Method

Kernel density analysis is a non-parametric method used to estimate the spatial distribution of point elements. It calculates the density of geographic elements within a certain range and determines the agglomeration area of their distribution. In this paper, we use kernel density analysis to measure the dispersion or agglomeration characteristics of the spatial distribution of National Archaeological Site Parks in China. The kernel density indicator can be calculated using the following formula [38]:

$$\hat{\lambda}_h(S) = \sum_{i=1}^{n} \frac{3}{\pi h^4} \left(1 - \frac{(S - S_i)^2}{h^2}\right)^2$$

where $S$ is the position of the object being estimated and $S_i$ is the position of the ith estimate object in a circle, with S as the center and h as the radius.

2.3.3. Standard Deviation Ellipse

The standard deviation ellipse is a tool for describing and explaining the centrality, directionality, and spatial distribution of geographical elements. Standard deviation changes in the major axis, minor axis, area of the ellipse, and center of gravity of the ellipse are used

to analyze the distribution characteristics, degree of aggregation, agglomeration center, and other spatial–temporal evolution rules of China's National Archaeological Site Parks and visually display their trends of spatial–temporal evolution and central directional migration. The standard elliptic difference parameters can be calculated using the following formula [39]:

$$SDE_x = \sqrt{\frac{\sum_{i=1}^{n}\left(x_i - \overline{X}\right)^2}{n}}, SDE_y = \sqrt{\frac{\sum_{i=1}^{n}\left(y_i - \overline{Y}\right)^2}{n}}$$

where $SDE_x$ and $SDE_y$ are the axis lengths of the standard deviation ellipse in the x and y directions, respectively. The major axis is the direction with the greatest spatial distribution, and the minor axis is the direction with the least spatial distribution. $x_i$ and $y_i$ are the coordinates of the locations of the National Archaeological Site Parks; $(\overline{X}, \overline{Y})$ is the average center of the spatial distribution of the National Archaeological Site Parks; and n is the total number of National Archaeological Site Parks.

### 2.3.4. Construction of the Indicator System

Through field research in National Archaeological Site Parks such as Beijing Yuanmingyuan, Henan Yinxu, and Sichuan Sanxingdui, and by interviewing experts, the development status, geographical locations, and surrounding environments of National Archaeological Site Parks were discerned and the factors influencing the development of local resources, the environment, and the surrounding society were selected. Referring to policy information such as the Rules for the Evaluation of National Archaeological Site Parks [40] and the Measures for the Administration of National Archaeological Site Parks [35], as well as Chinese scholars' constructions of indicators affecting the spatial–temporal patterns of cultural heritage, such as traditional ancient villages [41], key cultural relic protection units [42], intangible cultural heritage [43], etc., the influencing factors were constructed (Table 1).

**Table 1.** Indicator variables and descriptions.

| Impact Factors | Variable Name | Variable Meaning | Unit |
| --- | --- | --- | --- |
| Resources and environment | Large archaeological sites ($X_1$) | Number of large heritage sites per province | Number |
| | National key cultural relic protection units ($X_2$) | Number of national key cultural relic protection units per province | Number |
| | National A-class scenic spots ($X_3$) | Number of national A-class tourist attractions per province | Number |
| | World Cultural Heritage Sites ($X_4$) | Number of World Heritage Sites per province | Number |
| | The number of policies ($X_5$) | Number of national and local policies per province | Number |
| Social development | Population ($X_6$) | Number of people in towns and villages per province | 10 thousand people |
| | Total GDP ($X_7$) | The final output value of production activities of all permanent units per province | CNY 100 millions |
| | The output value of secondary industry ($X_8$) | The annual value of manufacturing, construction, and other industries' outputs per province | CNY 100 millions |
| | The output value of tertiary industry ($X_9$) | Annual value of the service sector's output per province | CNY 100 millions |
| | Tourist arrivals ($X_{10}$) | The number of tourist visits in a year per province | 10 thousand people |
| | Total tourism revenue ($X_{11}$) | The gross annual tourism revenue per province | CNY 100 millions |
| | Highway mileage ($X_{12}$) | The actual length of roads at the end of the year per province | Kilometers |

### 2.3.5. SPSS Factor Analysis

The principal component analysis using SPSS25.0 software was used to identify the relevant determinants and factors in order to construct the mathematical model for factor analysis. Firstly, using the idea of dimensionality reduction to decrease dependence within the correlation matrix of the original variables, some variables with intricate relationships were reduced to a few comprehensive factors [44]. Secondly, the significance of the main extracted factors was determined to quantify the role of each influencing factor.

## 3. Results

ArcGIS10.8 and EXCEL2016 software was used in this study to conduct statistical and spatial analyses on the batch, type, age, and spatial information of 135 National

Archaeological Site Parks in China. Three types of results were obtained: results on the time evolution characteristics of the National Archaeological Site Parks, results on the spatial evolution characteristics of the National Archaeological Site Parks, and results on the spatial–temporal evolution characteristics of the National Archaeological Site Parks. Moreover, SPSS25.0 software was used to construct the National Archaeological Site Parks and to perform factor analyses of the impact indicators, and two types of results were obtained: the identification of the principal components, and the quantitative scores of the impact factors of the spatial differentiation of the National Archaeological Site Parks.

### 3.1. Analysis of the Time Evolution of National Archaeological Site Parks

Using the statistical data of the National Archaeological Site Parks from 2010 to 2022, the statistical analysis function of EXCEL was employed to organize and output information on the age, type, and quantity of the large heritage sites upon which the National Archaeological Site Parks were established. Analytical results on the time evolution characteristics were obtained.

### 3.1.1. Changes in Age

Through the analysis of the temporal changes (Figure 4a), two major characteristics were found: Firstly, the large heritage sites selected as the National Archaeological Site Parks are relatively balanced in terms of their age, and various historical periods are represented. This is due to the continuous adjustment of the selection guide. Taking the first batch of National Archaeological Site Parks as an example, most of them date back to before the Tang Dynasty. In subsequent evaluations of the National Archaeological Site Parks, the imbalanced age distribution was clearly recognized, and more sites from the Liao, Song, Jin, Yuan, Ming, and Qing dynasties were selected as potential sites. Secondly, more attention has been paid to sites spanning longer time periods, such as the original Sanxingdui site in Sichuan, the Jingdezhen Imperial Kiln, the Jizhou Kiln, and other sites spanning multiple eras.

### 3.1.2. Changes in Type

In terms of heritage types (Figure 4b), the National Archaeological Site Parks cover various types of large heritage sites. Examples include prehistoric settlement sites that bear the origins of Chinese civilization; ancient urban sites that reflect the evolution of China's history, changes in social systems, and economic and cultural development; tombs and royal tombs that embody the ideas and ritual systems of the ancient Chinese ruling class; handicraft sites that reflect the peak of Chinese traditional crafts and the circulation of a commodity economy; etc. The sites themselves are more generally typical representations of national significance, historical significance, and cultural significance.

Further comparison of the quantity of each type of site revealed the following characteristics: Firstly, the National Archaeological Site Parks mainly consist of settlement sites and urban sites, with a total of 90 projects (including approved projects) in these two categories, accounting for 66.7% of the total. Secondly, handicraft sites, especially kiln sites, have suddenly risen in number, rapidly growing from 0 projects in the first batch to 14 projects (including approved projects). Thirdly, large-scale engineering sites have also begun to emerge, such as the Grand Canal Nanwang Hub site selected in the second batch, which has set the tone for the diversification of types of National Archaeological Sites Parks in the future.

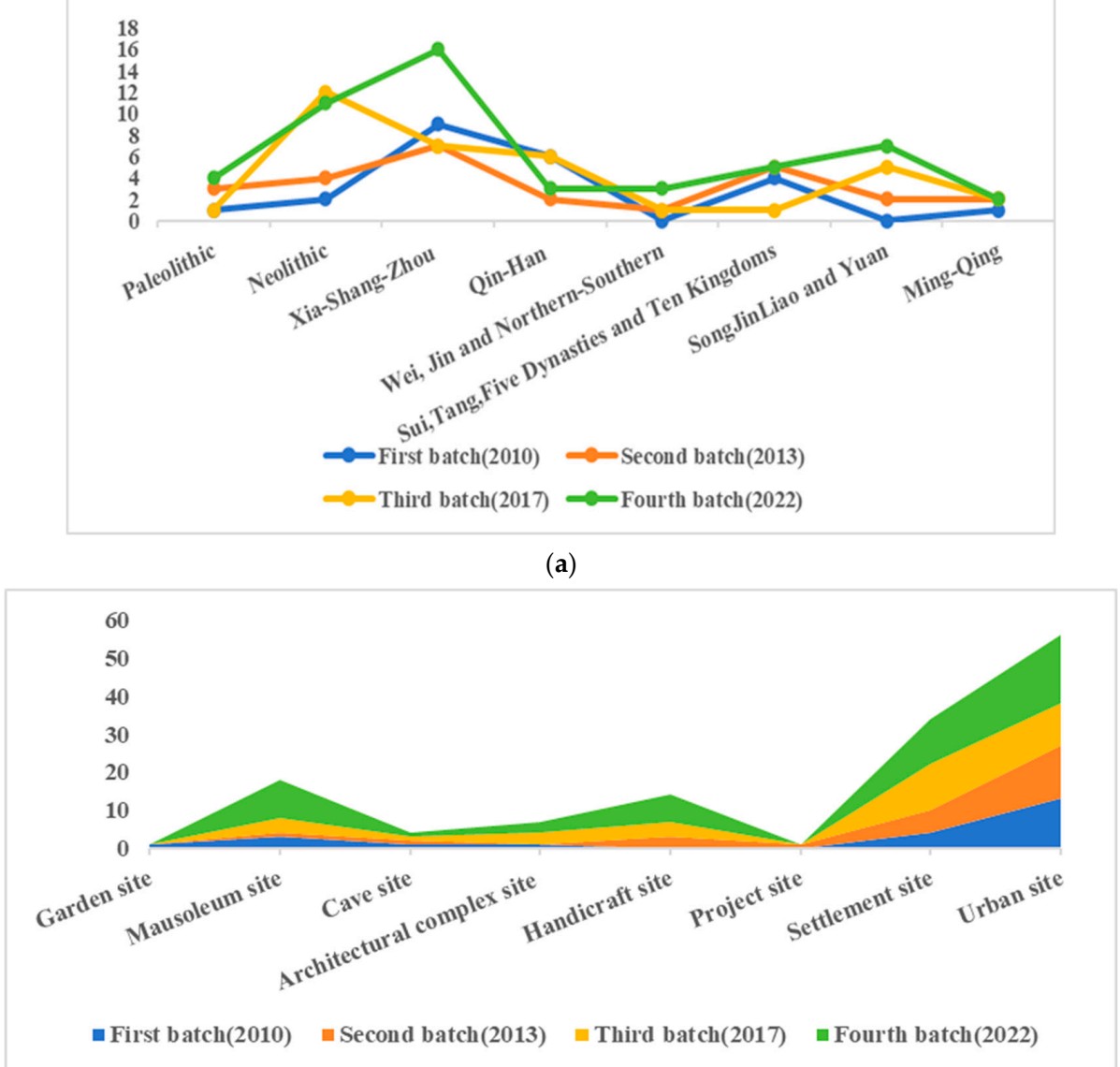

**Figure 4.** Changes in the ages and types of National Archaeological Site Parks between 2010 and 2022. (**a**) Changes in age. (**b**) Changes in type.

*3.2. Analysis of the Spatial Evolution of National Archaeological Site Parks*

Using statistical data on China's National Archaeological Site Parks from 2010 to 2022, the statistical analysis functionality of EXCEL was employed to organize and output information on the areas in which the National Archaeological Site Parks are located and to obtain the distribution characteristics of the National Archaeological Site Parks in terms of their provinces and types. Moreover, nearest-neighbor indicator calculation, kernel density calculation, and image visualization were conducted using the spatial information of the National Archaeological Site Parks in ArcGIS10.8 software to obtain their overall distribution characteristics.

3.2.1. Provincial Distribution Characteristics

There are National Archaeological Site Parks distributed across 27 provincial-level administrative regions in China (including the 34 regions in total), accounting for about 79.41% of all of the provinces. Among them, Henan, Shaanxi, Hubei, Zhejiang, and Hebei

have 17, 15, 11, 8, and 7 National Archaeological Site Parks, respectively; Sichuan, Hunan, Shandong, and Anhui each have 6 parks; Jilin, Jiangsu, Jiangxi, and Fujian each have 5 parks; and the remaining regions each have less than 5 parks (Figure 5). It can be seen from these findings that the core concentration area of the National Archeological Site Parks follows the spatial layout of the two core areas of Chinese civilization in the middle and lower reaches of the Yellow River and the middle and lower reaches of the Yangtze River. Henan and Shaanxi are attributed as the birthplaces of Chinese civilization and the Chinese nation. With their rich historical and cultural heritage and many historical relics, these two provinces have the largest number of National Archaeological Site Parks.

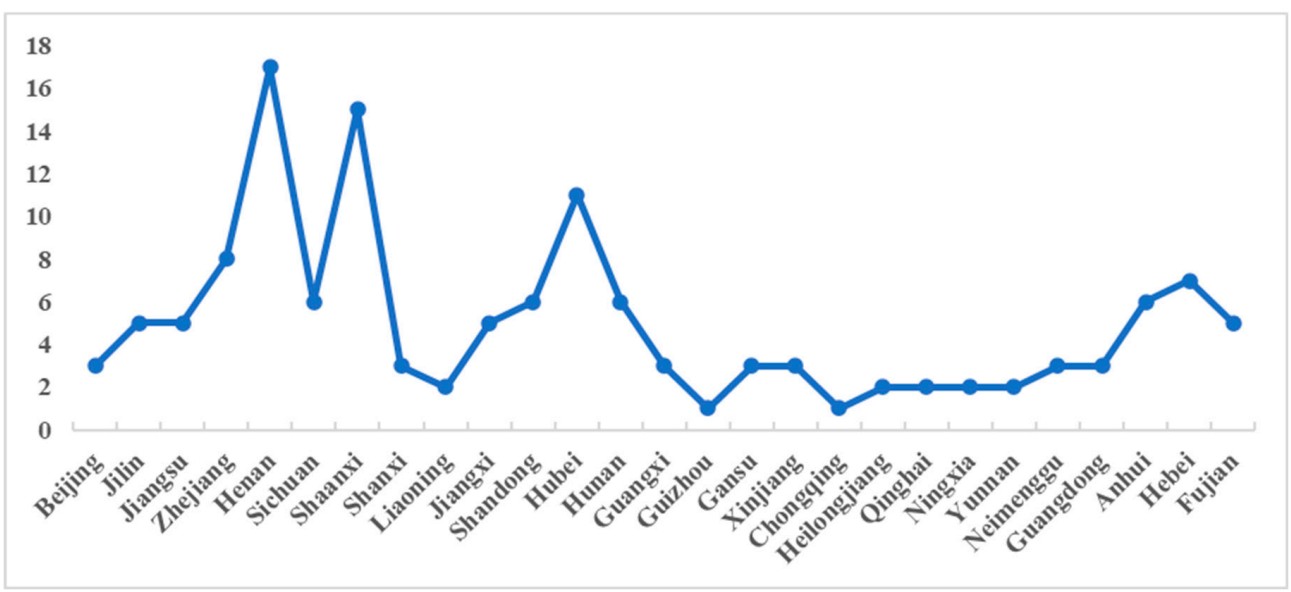

**Figure 5.** Number of National Archaeological Site Parks in 27 regions of China in 2022.

### 3.2.2. Type Distribution Characteristics

Encompassing a variety of large heritage site entities, National Archaeological Site Parks are established based on the information regarding the development of such sites officially released by the State Administration of Cultural Heritage of China on 18 April 2022 at the International Day of Monuments and Sites event [36]. As shown in Figure 6a, urban sites, settlement sites, and mausoleum sites account for 41.48%, 25.19%, and 13.33% of all National Archaeological Site Parks, respectively, and other types account for relatively small proportions. Figure 6b shows the distribution of National Archaeological Site Parks in various regions. (1) There is a wide variety of site types distributed in Central China, mainly including urban sites, mausoleum sites, and settlement sites. Sites in this area mostly reflect China's historical evolution, changes to the social system, and economic and cultural development. (2) East China is the epicenter of traditional Chinese crafts and the main area of the circulating commodity economy in Chinese history, and as such, it contains many handicraft sites. (3) Complex architectural sites are mostly distributed in the northwest region, including palaces from ancient dynasties, temple sites, and Buddhist temple sites. (4) The types of sites distributed in North China include urban sites, garden sites, and engineering sites. Generally speaking, China's National Archaeological Site Parks are mainly concentrated in East China, Central China, and North China, while the northeast, south, and southwest regions have fewer sites with a smaller variety of types.

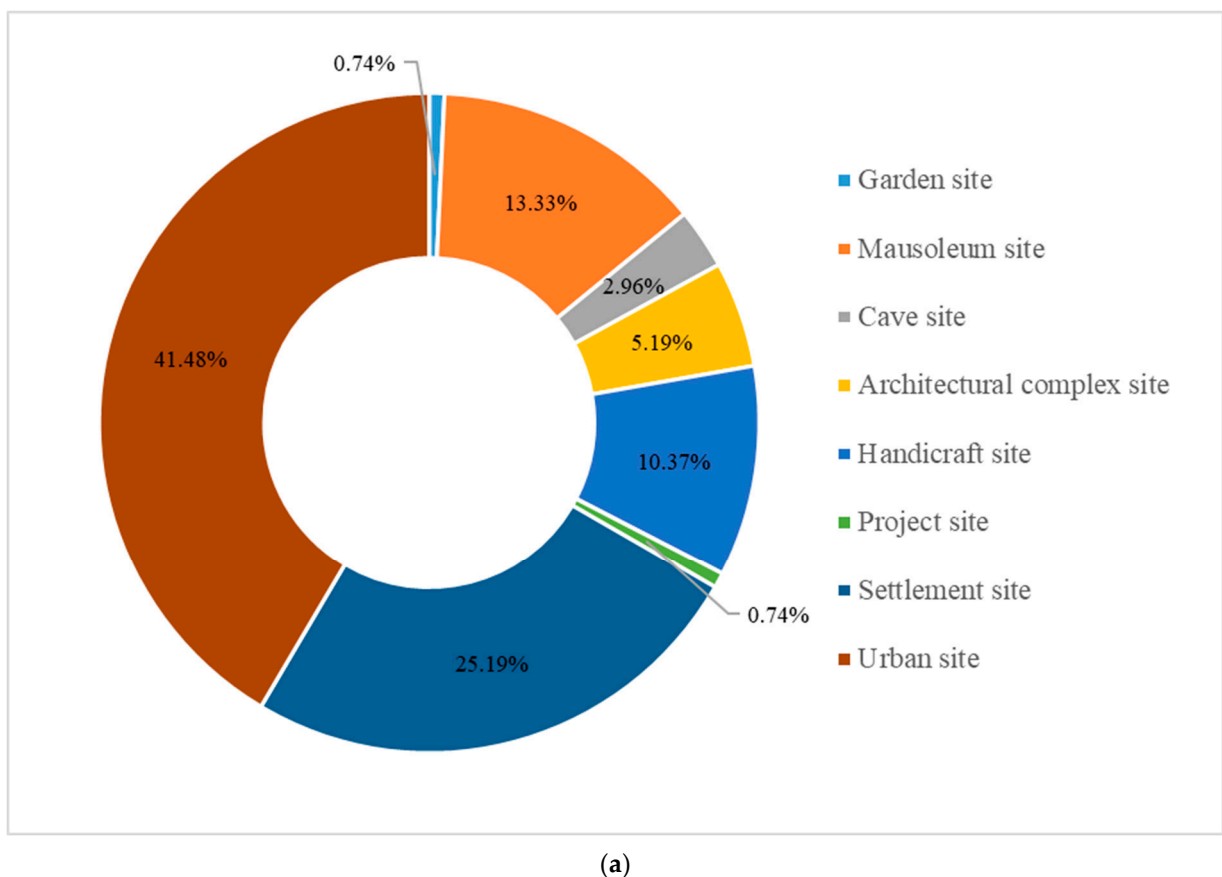

(**a**)

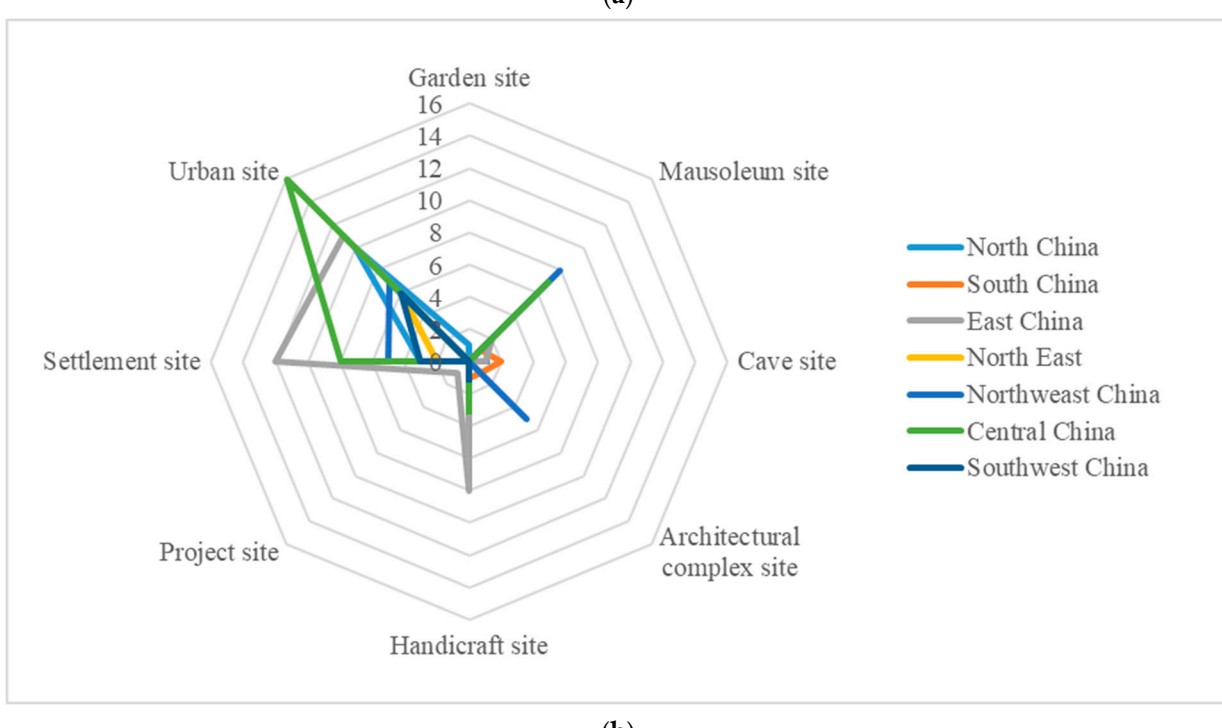

(**b**)

**Figure 6.** Site types of National Archaeological Site Parks and their distribution in seven regions of China. (**a**) Site types. (**b**) Distribution of different site types across seven major regions of China.

### 3.2.3. Overall Distribution Characteristics

Analyzing the spatial distribution characteristics of the National Archaeological Site Parks in Figure 7, it can be seen that most National Archaeological Site Parks are distributed in East and Central China, forming a spatial distribution pattern that is dense in the east and sparse in the west, which is basically consistent with China's "Hu Line" (Heihe City, Heilongjiang province, China (127.528° E, 50.245° N) to Tengchong City, Yunnan province (98.490° E, 25.020° N)) [45]. The nearest-neighbor indicator was used to determine the type of spatial distribution of the National Archaeological Site Parks, and the average nearest-neighbor distance of the National Archaeological Site Parks is 98.96 km, as calculated using ArcGIS10.8; when this is compared with the theoretical nearest-neighbor distance of 168.55 km, the nearest-neighbor indicator is 0.59 < 1, indicating that the National Archaeological Site Parks have a clustered spatial distribution trend.

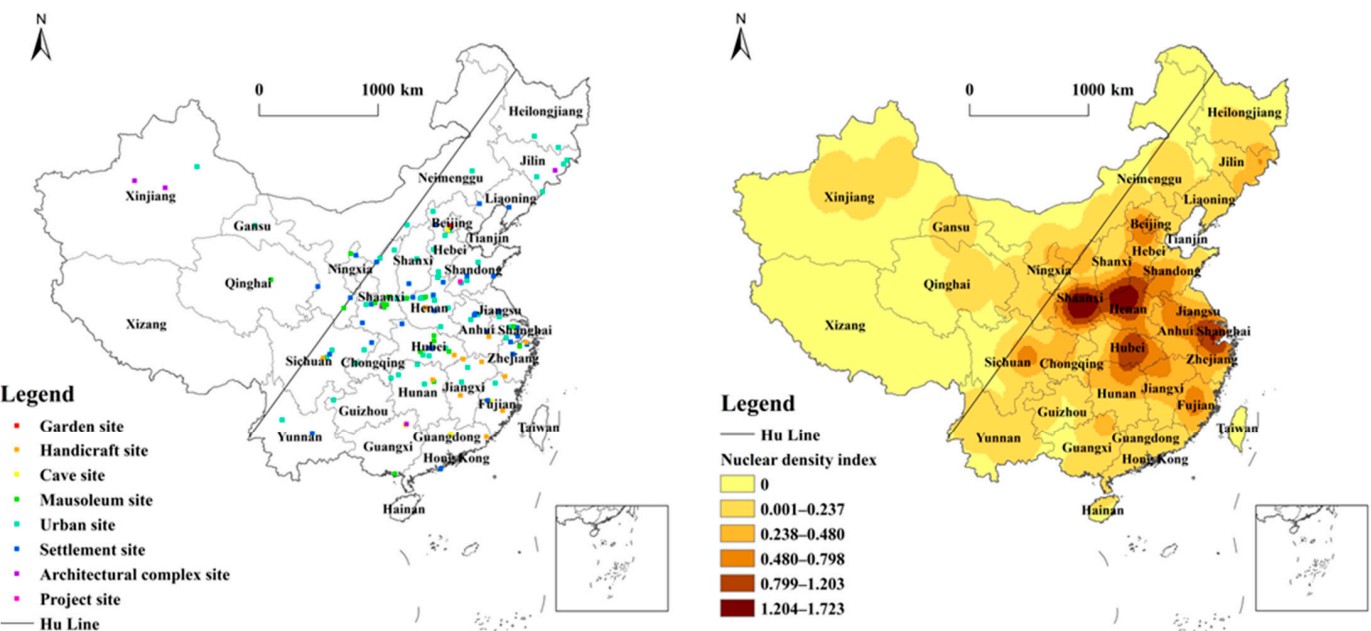

**Figure 7.** Spatial distribution of kernel density across the National Archaeological Site Parks.

After further calculations and analyses, it was found that the eastern, central, western, and northeastern regions also show agglomeration trends. ArcGIS10.8 software was used to conduct kernel density analysis on the National Archaeological Site Parks based on the quantile classification rules and to analyze the probability of the occurrence of point elements of National Archaeological Site Parks in different spaces. The spatial distribution of the National Archaeological Site Parks forms a high-density area in Central China, dominated by the Henan province and Hubei province, and a high-density area in East China, dominated by the Shandong province, Zhejiang province, and Anhui province. Meanwhile, the North China region, dominated by the provinces of Beijing–Tianjin–Hebei and Shanxi; the Northeast region, dominated by the Jilin province; and the eastern Southwest region, dominated by Sichuan and Chongqing, are sub-high-density areas, and the South and Northwest regions are low-density areas.

### 3.3. Analysis of the Spatial–Temporal Evolution of National Archaeological Site Parks

ArcGIS10.8 software was used to draw the SDE, taking the number of National Archaeological Site Parks in each province (city, district) in China per batch as the weight. The aim was to display the major regional changes in the spatial distribution of National Archaeological Site Parks and calculate the elliptic range and the center of gravity coordinates of their spatial distribution. The spatial–temporal evolution characteristics were obtained through analyses of these results.

### 3.3.1. Diffusion of the Spatial–Temporal Patterns

The following findings are demonstrated in Figure 8: (1) From 2005 to 2010, the first batch of National Archaeological Site Parks (including approved projects) were mainly concentrated in the Henan, Shaanxi, and Shandong provinces in Central China. As important birthplaces of Chinese civilization, these areas have profound historical and cultural connotations and rich heritage resources, and thus have laid the foundation for the construction of National Archaeological Site Parks in the country. (2) From 2011 to 2015, the new National Archaeological Sites Parks selected in the second batch (including approved projects) mainly included central cities in the Hebei, Ningxia, Hunan, and Jiangxi provinces, and on the Liaodong Peninsula. These sites are mainly located in the centers of cities, gradually diverging to the north, northwest, and southwest. (3) From 2016 to 2021, the third batch of National Archaeological Sites Parks (including approved projects) was distributed in a wide range from Xinjiang in the northwest to Heilongjiang in the northeast and Guangxi in the southwest. (4) From 2022 to the present, the number National Archaeological Site Parks selected in the fourth batch (including approved projects) increased significantly in various regions and now encompasses a larger scale. The sites in this batch are widely distributed in urban centers, suburbs, and villages across 27 provinces.

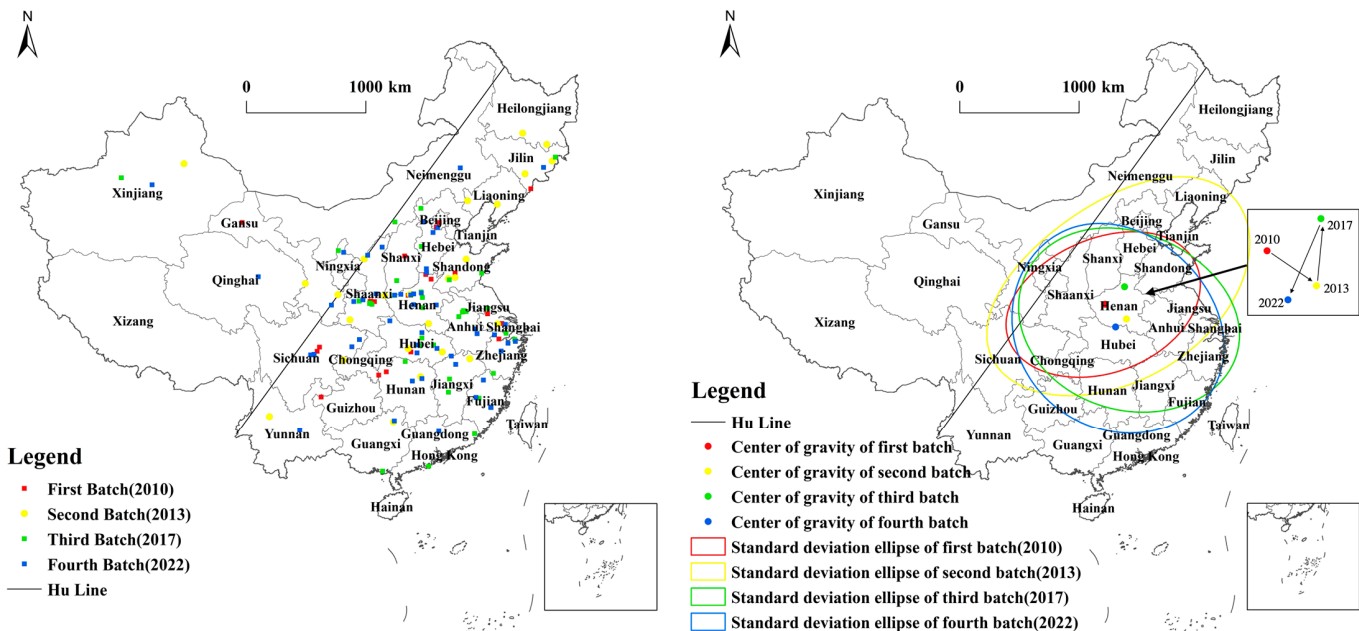

**Figure 8.** Spatial distribution and standard deviation ellipses of different batches of National Archaeological Site Parks.

### 3.3.2. Spatial Pattern Evolution

Standard deviation ellipse quantification is adopted in this paper to dynamically reveal the spatial–temporal evolution of the different batches of National Archaeological Site Parks. The following can be seen from the standard deviation ellipse parameters in Table 2: (1) The central coordinates of the standard deviation ellipse of the National Archaeological Site Parks in 2010 were located in Luoyang city, Henan province (111.942° E, 34.110° N). In 2013, these coordinates moved slightly northeast to Xinxiang city, Henan province (113.924° E, 35.298° N); in 2017, they moved southwest to Zhumadian city, Henan province (113.764° E, 32.823° N); and in 2022, they moved southwest to Xiangyang city, Hubei province (112.713° E, 32.303° N), which shows that the distribution of the National Archaeological Site Parks has experienced a shifting trend from the northeast to the southwest. (2) The elliptical area first increased, then decreased, and then increased again, indicating that the spatial patterns of the National Archaeological Site Parks have experi-

enced aggregation, diffusion, and then agglomeration again, and the overall pattern shows a process of dynamic adjustment and improvement. (3) The gap between the major and minor semi-axes of the standard deviation ellipse has narrowed overall, indicating that the directionality of the spatial distribution of the National Archaeological Site Parks is becoming weaker and the equilibrium distribution trend is strengthening.

**Table 2.** Standard deviation ellipse parameters for the four batches of National Archaeological Site Parks.

| Year | Central Coordinates | XStdDist/km | YStdDist/km | Rotation/° | Direction of Movement | Area/km$^2$ |
|------|---------------------|-------------|-------------|------------|-----------------------|-------------|
| 2010 | 111.942° E, 34.110° N | 9.23 | 5.20 | 78.61 | Northeast | 1,558,514.90 |
| 2013 | 113.924° E, 35.298° N | 13.32 | 7.04 | 70.90 | Northeast | 2,993,628.97 |
| 2017 | 113.764° E, 32.823° N | 10.01 | 6.92 | 105.93 | Southwest | 2,272,225.18 |
| 2022 | 112.713° E, 32.303° N | 9.60 | 7.83 | 112.66 | Southwest | 2,479,470.16 |

*3.4. Analysis of Factors Influencing the Spatial Distribution of the National Archaeological Site Parks*

With the help of the factor analysis function in SPSS25.0, correlation testing, principal components identification, and factor analysis of the influencing factor data were performed; EXCEL2016 was used to calculate and collate the results of the principal component identification of the influencing factors of the National Archaeological Site Parks of China and the results of quantitative scores of the influencing factors.

3.4.1. Correlation Test of Influencing Factors

Correlation testing of the influencing factors using SPSS25.0 software shows that the selected indicator variables are correlated with each other: the KMO test statistic is 0.686, the Bartlett's test statistic is 286.537, and the associated probability is 0, meaning that the influencing factors are suitable for factor analysis.

3.4.2. Principal Component Analysis Results for the Influencing Factors

To make the meaning of the principal components clearer, as shown in Table 3, the factor-loading matrix was rotated according to the Varimax method. In this method, four principal components with eigenvalues greater than 1 were extracted, and the cumulative variance contribution rate reached 83.928%. This finding shows that these four principal components comprehensively summarize the meaning of all twelve indicators and can properly measure the four factors affecting the distribution characteristics of the four batches of National Archaeological Site Parks in the country of China.

**Table 3.** Principal component variance contribution analysis results for the influencing factors of the National Archaeological Site Parks extracted using the maximum-variance method.

| Principal Component | % of Variance (Rotated) | | |
|---------------------|-------------|----------------|-----------------------------|
| | Eigenvalue | % of Variance | Cumulative % of Variance |
| $F_1$ | 3.945 | 32.879 | 32.879 |
| $F_2$ | 2.552 | 21.269 | 54.148 |
| $F_3$ | 2.246 | 18.716 | 72.863 |
| $F_4$ | 1.328 | 11.064 | 83.928 |

The identified principal components were analyzed based on the literature reviewed and the construction and development of the National Archaeological Site Parks. These results are shown in Table 4.

Component $F_1$ has four indicators: total GDP, value of tertiary industry output, value of secondary industry output, and regional population. This principal component represents the "regional development" factor, with a contribution rate of 32.879%. It is the largest

influencing factor in the construction of the National Archaeological Site Parks, reflecting the influence of external driving forces.

Component $F_2$ includes four indicators: number of tourists, tourism revenue, highway mileage, and the number of national A-class scenic spots. This principal component measures the driving influence of regional heritage tourism on the construction of National Archaeological Site Parks, and serves as the second most influential factor affecting the distribution of National Archaeological Site Parks.

Component $F_3$ includes indicators such as the number of large heritage sites, the number of national key cultural relic protection units, and the number of World Cultural Heritage Sites. This principal component represents historical resources, with a contribution rate of 18.716%. As the third major influencing factor, it reflects the endogenous driving force of the construction of National Archaeological Site Parks.

Component $F_4$ represents the policy support capacity, reflecting the impact of government planning and support on the construction of the National Archaeological Site Parks. It serves as another important factor affecting the spatial distribution of the archaeological site parks.

**Table 4.** Correlation matrix of the 4 influencing factors and 12 indicators measured after variance maximization rotation.

| Indicator | Principal Component | | | |
|---|---|---|---|---|
| | F1 (Regional Development Factor) | F2 (Heritage Tourism Factor) | F3 (Historical Resource Factor) | F4 (Government Support Factor) |
| Total GDP | 0.953 | 0.183 | 0.14 | 0.044 |
| The value of the tertiary industry output | 0.95 | 0.144 | 0.126 | 0.043 |
| The value of the secondary industry output | 0.89 | 0.21 | 0.157 | 0.136 |
| Population | 0.828 | 0.419 | 0.225 | −0.076 |
| Tourist arrivals | 0.183 | 0.86 | 0.017 | 0.2 |
| Highway mileage | 0.199 | 0.773 | 0.136 | −0.317 |
| Total tourism revenue | 0.551 | 0.671 | 0.304 | 0.018 |
| Number of national A-class scenic spots | 0.453 | 0.632 | 0.165 | 0.479 |
| Number of World Cultural Heritage Sites | 0.101 | −0.07 | 0.835 | 0.014 |
| Number of national key cultural relic protection units | 0.198 | 0.184 | 0.833 | 0.065 |
| Number of large-scale archaeological sites | 0.158 | 0.233 | 0.761 | 0.189 |
| Number of policies | 0.042 | −0.003 | 0.165 | 0.943 |

### 3.4.3. Quantitative Scoring Results for the Influencing Factors

In order to obtain the scores of factors influencing the spatial differentiation of National Archaeological Site Parks in different regions, it is necessary to calculate the coefficient of the common factor score. The regression method was used to obtain the factor component score coefficient matrix, as shown in Table 5.

**Table 5.** Score coefficient matrix of the four influencing factors of the National Archaeological Site Parks.

| Indicator | Component | | | |
|---|---|---|---|---|
| | $F_1$ (Regional Development Factor) | F2 (Heritage Tourism Factor) | F3 (Historical Resource Factor) | $F_4$ (Government Support Factor) |
| Number of large-scale archaeological sites | −0.094 | 0.029 | 0.37 | 0.047 |
| Number of national key cultural relic protection units | −0.068 | −0.015 | 0.426 | −0.062 |
| Number of national A-class scenic spots | −0.013 | 0.243 | −0.077 | 0.334 |
| Number of World Cultural Heritage Sites | −0.04 | −0.146 | 0.468 | −0.093 |
| Number of policies | −0.035 | −0.055 | −0.027 | 0.742 |
| Population | 0.213 | 0.023 | 0 | −0.131 |
| Total GDP | 0.334 | −0.152 | −0.057 | −0.021 |
| The value of secondary industry output | 0.295 | −0.123 | −0.054 | 0.054 |
| The value of tertiary industry output | 0.344 | −0.174 | −0.062 | −0.018 |
| Tourist arrivals | −0.159 | 0.478 | −0.115 | 0.129 |
| Total tourism revenue | 0.017 | 0.243 | 0.043 | −0.062 |
| Highway mileage | −0.125 | 0.423 | 0.022 | −0.303 |

Standardizing the original indicator value of each sample, the standardized score of each indicator and the component score coefficient matrix was used to calculate the common factor score of each sample, using the following formula:

$$F_{ij} = \sum P_{ik} Q_{kj} \tag{1}$$

where $F_{ij}$ represents the ith common factor score of the jth sample, $P_{ik}$ is the component score coefficient of the ith common factor on the kth indicator, and $Q_{kj}$ is the kth indicator score of the jth sample after standardization. The equation calculating the score of the three common factors is expressed as follows:

$$F_1 = -0.094X_1 - 0.068X_2 - 0.013X_3 + 0.040X_4 - 0.035X_5 + 0.213X_6 + 0.334X_7 + 0.295X_8 + 0.344X_9 - 0.159X_{10} + 0.017X_{11} - 0.125 \tag{2}$$

$$F_2 = 0.029X_1 - 0.015X_2 + 0.243X_3 - 0.146X_4 - 0.055X_5 + 0.023X_6 - 0.152X_7 - 0.123X_8 - 0.174X_9 + 0.478X_{10} + 0.243X_{11} + 0.423X_{12} \tag{3}$$

$$F_3 = 0.370X_1 + 0.426X_2 - 0.077X_3 + 0.468X_4 - 0.027X_5 - 0.057X_7 - 0.054X_8 - 0.062X_9 - 0.115X_{10} + 0.043X_{11} + 0.022X_{12} \tag{4}$$

$$F_4 = 0.047X_1 - 0.062X_2 + 0.334X_3 - 0.093X_4 + 0.742X_5 - 0.131X_6 - 0.021X_7 + 0.054X_8 - 0.018X_9 + 0.129X_{10} - 0.062X_{11} - 0.303X_{12} \tag{5}$$

The common factor score is weighted according to the variance contribution rate after rotation, and the comprehensive score F is obtained:

$$F = \frac{W_1 F_1 + W_2 F_2 + W_3 F_3 + W_4 F_4}{W_1 + W_2 + W_3 + W_4} \tag{6}$$

where $W_i$ is the variance contribution rate after rotation.

The scores and rankings of the four common factors in the 27 regions can be calculated using Formulas (2)–(5). The variance contribution rate and cumulative variance contribution rate of the four main indicators in Table 3 are substituted into Formula (6) and then the comprehensive scores and rankings of the 27 regions are obtained for the four factors, as shown in Table 6: urban development, heritage tourism, historical resources, and government support. There are significant differences in the scores of regional development, heritage tourism, historical resources, and government support across the different regions in China, which profoundly affects the degree of spatial differentiation of the National Archaeological Site Parks.

**Table 6.** Influencing factors' scores and rankings of National Archaeological Site Parks in 27 regions of China.

| Province | Aggregate Score | Aggregate Ranking | F₁ (Regional Development Factor) | F₂ (Heritage Tourism Factor) | F₃ (Historical Resource Factor) | F₄ (Government Support Factor) |
|---|---|---|---|---|---|---|
| Zhejiang | 1.073 | 1 | 0.853 | 0.478 | 0.089 | 4.537 |
| Shandong | 0.937 | 2 | 1.298 | 1.230 | 0.586 | −0.106 |
| Henan | 0.822 | 3 | 0.257 | 0.507 | 2.831 | −0.294 |
| Guangdong | 0.709 | 4 | 3.581 | −1.280 | −1.270 | −0.652 |
| Jiangsu | 0.657 | 5 | 1.802 | −0.189 | 0.223 | −0.382 |
| Sichuan | 0.487 | 6 | 0.223 | 1.792 | 0.033 | −0.469 |
| Hubei | 0.426 | 7 | 0.789 | 0.731 | −0.130 | −0.296 |
| Shaanxi | 0.190 | 8 | −0.682 | 0.166 | 1.421 | 0.743 |
| Anhui | 0.163 | 9 | 0.020 | 0.733 | 0.064 | −0.338 |
| Hebei | 0.135 | 10 | 0.036 | −0.123 | 1.036 | −0.598 |
| Hunan | 0.098 | 11 | 0.258 | 0.658 | −0.460 | −0.508 |
| Fujian | 0.007 | 12 | 0.442 | −0.870 | 0.150 | 0.157 |
| Yunnan | −0.061 | 13 | −0.842 | 1.916 | −0.696 | −0.466 |
| Jiangxi | −0.106 | 14 | −0.487 | 0.776 | −0.254 | −0.418 |
| Shanxi | −0.118 | 15 | −0.465 | −1.077 | 1.976 | −0.786 |
| Liaoning | −0.163 | 16 | −0.238 | −0.291 | 0.176 | −0.264 |
| Guangxi | −0.166 | 17 | −0.573 | 1.263 | −1.122 | −0.088 |
| Beijing | −0.180 | 18 | −0.147 | −1.145 | 1.046 | −0.494 |
| Guizhou | −0.286 | 19 | −0.666 | 0.951 | −1.112 | −0.133 |
| Neimenggu | −0.393 | 20 | −0.580 | −0.664 | −0.261 | 0.461 |
| Xinjiang | −0.397 | 21 | −0.849 | −0.262 | 0.188 | −0.301 |
| Chongqing | −0.399 | 22 | −0.263 | 0.126 | −1.203 | −0.451 |
| Gansu | −0.407 | 23 | −0.906 | −0.730 | −0.004 | 1.017 |
| Jilin | −0.575 | 24 | −0.695 | −1.078 | −0.383 | 0.424 |
| Heilongjiang | −0.588 | 25 | −0.573 | −0.450 | −0.980 | −0.229 |
| Qinghai | −0.911 | 26 | −0.876 | −1.428 | −0.945 | 0.036 |
| Ningxia | −0.957 | 27 | −0.716 | −1.739 | −0.998 | −0.104 |

As shown in Figure 9, the Zhejiang, Shandong, and Henan provinces have higher scores in the regional development, heritage tourism, historical resources, and government support factors. These regions are particularly prominent in terms of their policy support factors and for their advantageous combination of heritage tourism and historical resource factors. Overall, the spatial differences between National Archaeological Site Parks in various regions of China are mainly reflected in the heritage tourism and historical resources factors, and regional economic development and policy support in a region are guarantors for the construction and development of National Archaeological Site Parks.

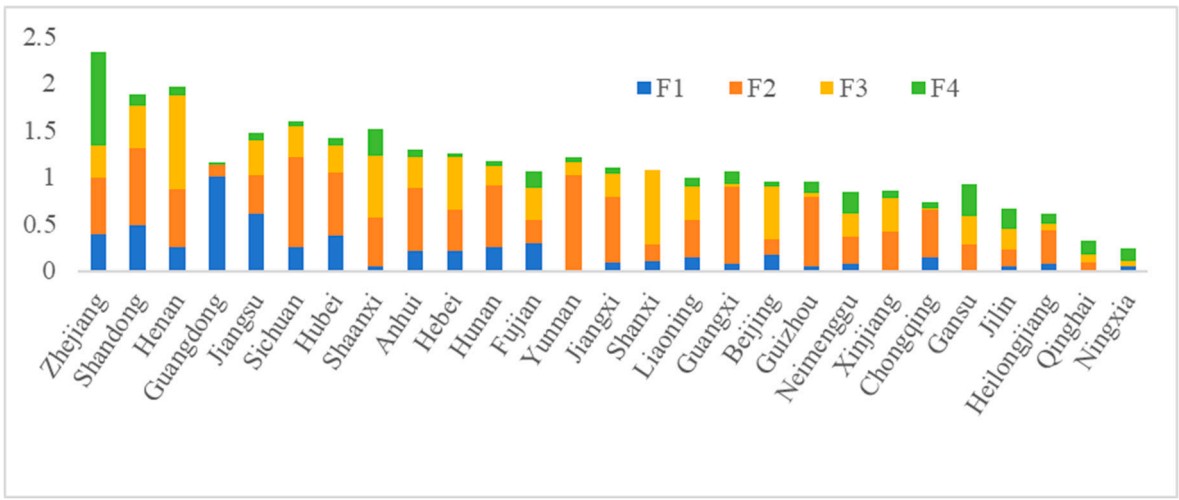

**Figure 9.** Results of the composition of scores for the influencing factors of National Archaeological Site Parks in 27 regions of China.

As shown in Figure 10, to facilitate further evaluations and comparisons between different regions, the scores of the regional development factor, heritage tourism factor, historical resource factor, and government support factor for the 27 regions in China with National Archeological Site Parks are analyzed separately.

(1)    Analysis of "regional development factors" in each region

The top-ranked regions for this factor are Guangdong and Jiangsu. This is because Guangdong and Jiangsu have developed economies and large populations, providing great external advantages for the operation and management of the National Archaeological Site Parks. The lowest scores for this factor are in Qinghai and Gansu. The weak economic foundations in these regions have a certain limiting effect on the construction and operation of National Archaeological Site Parks, which affects the degree of heritage protection and site distribution.

(2)    Analysis of "heritage tourism factors" in each region

The top two regions in terms of the heritage tourism factor are Yunnan and Sichuan, and the region with the lowest score is Ningxia. Yunnan and Sichuan are important provinces for tourism in China, and their high scores in this factor show that good tourism development can promote the inflow of cultural elements and archaeological site resources. The lack of a notable tourism benefit and the imperfect transportation facilities in Ningxia are the reasons why the scale of construction and quality of development of its National Archaeological Site Parks are limited.

(3)    Analysis of "historical resource factors" in each region

The regions with the highest scores for the historical resources factor are Henan and Shaanxi. Because some cities in Henan and Shaanxi served as capitals during different dynasties in Chinese history, these two provinces have a richer history and culture due to their many major sites and cultural relic protection units. This affects the functions of the

National Archaeological Site Parks and the direction of their cultural evolution. In Ningxia and Qinghai, due to their low abundance of historical and cultural resources, the spatial distribution of National Archaeological Site Parks is quite different from other regions.

(4)   Analysis of "government support factors" in each region

China shows relatively obvious policy-oriented characteristics in the assessment and construction of National Archaeological Site Parks. For example, the Zhejiang provincial government has optimized the spatial pattern of its National Archaeological Site Parks by coordinating the scale and quality of the National Archaeological Site Parks' development using legal, economic, or social coercive force. Provinces such as Shanxi and Guangdong should make full use of the government's attitude and policy and financial support to promote the development of National Archaeological Site Parks.

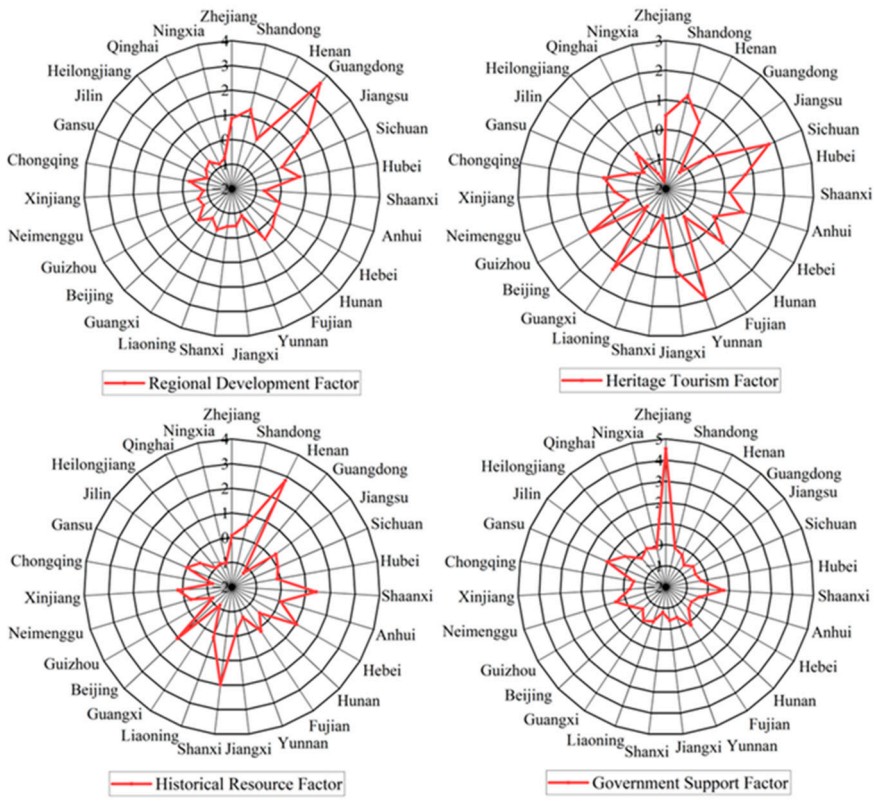

**Figure 10.** Scores of the four influencing factors for National Archaeological Site Parks in 27 regions of China.

## 4. Discussion

National Archaeological Site Parks are a strategic national project in China for innovatively exploring the scientific protection and rational utilization of large heritage sites with the aim of balancing urban development and protecting cultural heritage. In the context of urban renewal and heritage preservation, the active use of National Archaeological Site Parks can make sites a soft but effective governance tool in sustainable urban development. The study of the spatial–temporal evolution of National Archaeological Site Parks reveals a significant imbalance in their spatial patterns from 2010 to 2022, which is gradually decreasing. This is because China considers the balance and synergy of the national spatial layout when evaluating and constructing National Archaeological Site Parks, aiming to ensure the scientific protection and management of large heritage sites in each province (or its cities and districts) through the development of these sites. The Chinese government also aims to make archaeological achievements beneficial to the entire population. The results of the influencing factor identification found that in the construction process of China's

National Archaeological Site Parks, historical resource elements are foundational, the level of regional development is key, heritage tourism is the driving force, and the government's policy support is the guarantor of sustainable development. The coordination and linkage of these four major driving forces promote the construction and development of China's National Archaeological Site Parks. In this study, combined with the results of the analyses, the identified influencing factors are critically discussed, and specific strategies for promoting the rational utilization of large heritage sites and the sustainable development of National Archaeological Site Parks are proposed for each region.

### 4.1. Effects of Regional Development and Improvement Strategies

As products of the interaction between humans and the land throughout various historical periods, large heritage sites show a roughly positive correlation between the population and the spatial–temporal evolution of the National Archaeological Site Parks, and their overall distribution conforms to the "Hu Line" in China. Proposed by the famous Chinese geographer Mr. Hu Huanyong, the "Hu Line" is a geographical line dividing the population from Heihe city, Heilongjiang province (127.528° E, 50.245° N) to Tengchong city, Yunnan province (98.490° E, 25.020° N). Not only are there huge differences in population density between the two sides of the line, but there are also huge differences in economic and social development levels and development capabilities. From a local perspective, the three regions with the highest population densities in China (Beijing, Jiangsu, and Guangdong) have only 11 National Archaeological Site Parks. Economic development in these regions is rapid, and the combination of the scarcity of land for construction during early urbanization, the high demand for intensive land use, and the negative impacts of urbanization have increased the threats to monuments and heritage sites and their surroundings. This has also made it more challenging to establish National Archaeological Site Parks.

As the regional economy continues to grow, it is important to focus on improving industrial development and infrastructure in cities to support the construction of National Archaeological Site Parks. Additionally, measures should be taken to better protect these sites from urban expansion, allowing them to maintain their value while focusing on the cultural significance of the city. At the macro level, the Chinese government should adhere to the strategy of coordinated regional development and continue to accurately improve the infrastructure, the industrial layout, the investment landscape, etc., in the underdeveloped regions in the west so as to provide support for the coordination of the protection and utilization of large heritage sites. Managers of different regions should pay attention to the potential development conditions of archaeological site parks brought about by the population level and explore a sustainable model that coordinates regional economic development, the creation of urban habitats, and the functioning of National Archaeological Site Parks based on the consideration of the spatial and functional layout, land use planning, and so on [46].

### 4.2. Effects of Heritage Tourism and Improvement Strategies

The central and eastern parts of China have great historical resources and geographical advantages, numerous scenic tourist spots, and relatively mature tourism development. The Hubei province in Central China and the Fujian province in Eastern China each have more than 400 A-class scenic spots. The higher demand for cultural tourism and the benefits of a tourism sector promote the development of National Archaeological Site Parks. The number of National Archaeological Site Parks exceeds the number of large heritage sites, and most of the National Archaeological Site Parks are also national A-class tourist attractions. In recent years, with the development of tourist attractions and the improvement of transportation facilities in Western China, domestic tourism has developed rapidly. As a strategic pillar of industry that is highly adaptive and can trigger the redistribution of social resources and drive the economy, tourism provides a realistic possibility for the spatial–temporal patterns and center of gravity of China's National

Archaeological Site Parks' distribution to spread in the southwest direction. At present, the number of National Archaeological Site Parks in Western China exceeds 30% of that of the whole country.

It is worth pointing out that a sharp increase in the scale of heritage tourism can damage National Archaeological Site Parks. Therefore, tourism development must strictly adhere to the basic principle of not damaging the sites and environments on which they depend, and the capital, circulation, value orientation, and operation mode of heritage tourism development in National Archaeological Site Parks should give priority to heritage discourse. Tourism planning should take into account the coupling between the elements of the urban system and the archaeological site parks, rationally allocate resources by looking inward for value and uniqueness and outward for links with the surrounding environment, and promote the organic integration of the site's cultural offerings, tourism, and the city. Therefore, it is necessary to understand the heterogeneity of the types and spaces of different archaeological site parks; use scientific and technological innovation, digital drive, supply optimization, etc., to improve and create a sustainable tourism model; transform the site resources into tourist attractions to achieve sustainable economic development and drive comprehensive urban development; and inject new vitality into the role of the National Archaeological Site Parks to utilize and enhance their functions.

### 4.3. Effects of Historical Resources and Improvement Strategies

National Archaeological Site Parks are a management model for protecting large heritage sites in China, and their spatial distribution has a clear positive correlation with the distribution of heritage resources. Therefore, the rich historical resources and relatively large heritage sites in Central China and East China present a significant resource foundation for the construction of National Archaeological Site Parks. The Henan and Shaanxi provinces are the top two provinces in the country in terms of their numbers of large heritage sites and cultural relics protection units, and these provinces also have the largest numbers of National Archaeological Sites Parks. However, this positive correlation is not inevitable, as it is influenced by other factors. In the process of urban development, historical factors form the basis for the construction of National Archaeological Site Parks, and how to protect, utilize, and make full use of the social value of large sites is equally important. China's 14th Five-Year Plan for the Protection and Utilization of Large Sites [47] shows that there are nine large heritage sites in the Xinjiang Autonomous Region of China and eight large heritage sites in the Zhejiang province. The Zhejiang provincial government makes full use of its location, resources, and economic development advantages, and eight National Archaeological Site Parks have now been built. Meanwhile, there are only three sites in the Xinjiang Autonomous Region, and there is a need to focus on the history of the heritage resources and adjust the heritage protection plans to make these National Archaeological Site Parks a driving force and source of capital gains for the region's sustainable development.

National Archaeological Site Parks form a complex historical and cultural system composed of material remains and the surrounding natural and social environments. Managers should adaptively reuse archaeological site parks as a special social resource taking into full consideration the unique site characteristics and park attributes [48] so that the site parks can be fully integrated within urban public spaces, local communities, and the surrounding ecological environments. In this process, it is necessary to strengthen the excavation of regional culture; promote the integration and centralized and continuous protection and utilization of urban historical resources; further develop new industries, such as public cultural services, leisure tourism, and modern eco-agriculture, in relation to the sites, relying on different types of cultural relics and resources; and improve the patterns of the protection and utilization of National Archaeological Site Parks.

*4.4. Effects of Government Support and Improvement Strategies*

The establishment of National Archaeological Site Parks in China has significant externalities related to public welfare and the economy, and China's expenditure on cultural relic protection and infrastructure construction is its largest expenditure. It is necessary for the Chinese government to provide institutional and financial support through policies such as the 14th Five-Year Plan for the Protection and Utilization of Large Sites [47] and the Measures for the Management of Special Funds for the Protection of Major Sites [49], which have promoted the balanced development of the National Archaeological Site Parks. All aspects of National Archaeological Site Parks, from their declaration and establishment to their subsequent development, cannot be separated from the coordinating, linking, and supporting local policies. The level of political support reflects the importance that local governments attach to large heritage sites, which is also reflected in the spatial and developmental differentiation of National Archaeological Site Parks. The Zhejiang and Shaanxi provinces have adopted many policies to ensure the sustainable development of archaeological site parks, and their development of the tourism sector as well as the main function of these sites has achieved remarkable results. However, National Archaeological Site Parks from all four batches have problems such as poor financial expenditure channels and difficulties in starting the parks [50], which require local governments to tailor their policies to local conditions.

The Chinese government should clarify the main problems currently facing the protection and utilization of large heritage sites, continue to improve the management of large sites, and support the construction of National Archaeological Site Parks in terms of mechanisms, policy, and finances [51]. The Chinese government should also consider the variations in spatial elements and systems of National Archaeological Site Parks across all provinces, types, and locations and adopt diverse strategies for site protection and development. The management philosophy of National Archaeological Site Parks should be shifted from pure site protection to focusing on the sustainability of the site itself as well as the local economy, culture, and society, and promoting the coordinated development of the protection and utilization of large heritage sites and the local area. In addition to the unpredictability and spatial complexity of large sites, regional governments should integrate park construction into urban development plans from the perspective of spatial correlations and temporal dynamics, deepen the level of cooperation between various disciplines, and consider archaeological site parks as the "glue" between the different dimensions of sustainable development, so that they can bring broader and tangible benefits to local residents.

In summary, in the future, regional governments in China should pay attention to the influencing factors and mechanisms of National Archaeological Site Parks, and also to the harmonious relationship between regional sustainable development and the protection of large heritage sites.

## 5. Conclusions

This study reveals the spatial–temporal evolution patterns and influencing factors of China's National Archaeological Site Parks from 2010 to 2022 and proposes strategies to promote the rational utilization of large heritage sites and their sustainable development. In our analyses, the GIS analysis method was first used to quantitatively determine and visualize the spatial–temporal evolution patterns and distribution characteristics of the National Archaeological Site Parks. Secondly, two standard layers of resources and environmental factors, as well as social development factors, were identified through field research, expert interviews, and a literature review, and an indicator system of the factors influencing the spatial and temporal distribution of National Archaeological Site Parks was constructed. Factor analysis using SPSS was performed to estimate and identify the major driving factors. Finally, combining the results of the quantitative analysis, the effects of the influencing factors were discussed, and specific strategies to promote the rational utilization

of large heritage sites and the sustainable development of National Archaeological Site Parks were proposed.

Based on the innovative combination of "GIS analysis + construction of indicator system + SPSS factor analysis" and through the combined research path of "spatial–temporal evolution visualization + qualitative analysis of indicator system + quantitative analysis of influencing factors", five conclusions about the National Archaeological Site Parks are drawn:

(1)  In terms of the temporal evolution, the number of National Archaeological Site Parks in China increased from 2010 to 2022, and the large heritage sites gradually became more balanced in age and diversified in type.

(2)  In terms of spatial evolution, the Henan province and Shaanxi province are hotspots for China's National Archaeological Site Parks. The main concentration area of these sites closely follows the layout of China's two core areas, namely the middle and lower reaches of the Yellow River and the middle and lower reaches of the Yangtze River. Regionally, two high-density areas were identified in Central China and East China. The overall distribution conforms to the "Hu Line".

(3)  In terms of the spatial–temporal evolution, China's National Archaeological Site Parks have experienced trends of both diffusion and agglomeration, with the overall distribution range gradually expanding and the center of gravity of the trajectory gradually advancing toward Southwest China. Additionally, the equilibrium distribution trend has been increasing.

(4)  The spatial–temporal patterns of China's National Archaeological Site Parks are comprehensively affected by a variety of factors, among which historical resource elements are the foundation, the regional development level is key, heritage tourism development is the driving force, and government policy support is the guarantor of sustainable development. These four major driving forces jointly promote the establishment and development of National Archaeological Site Parks through coordination and linkage.

(5)  Governments in all regions of China should increase their efforts to protect their cultural heritage resources and combine the protection and utilization of sites with regional development strategies and urban renewal plans. Managers should actively develop culture and tourism industries in accordance with site characteristics and the surrounding ecological environments so as to achieve sustainable cultural, economic, ecological, and social development. In this way, National Archaeological Site Parks can be promoted as a new type of driving force for urban renewal.

The main contributions of the paper are as follows: Firstly, this paper expands the research on National Archaeological Site Parks by combining geographical and statistical perspectives, thus enriching the existing literature. It also complements the systematic and fundamental research work conducted on the issue of the spatial and temporal patterns of National Archaeological Site Parks on a large scale. Secondly, this paper uses GIS analysis, indicator system construction, and SPSS factor analysis to establish a method for studying the spatial–temporal patterns and influencing factors of National Archaeological Site Parks on a large scale, constructing a scientific research model that includes the "visualization of spatial–temporal evolution" and "qualitative analysis of indicator system" and "quantitative analysis of influencing factors". This approach not only overcomes the limitations of traditional, single-method research, but also provides a scientific and practical research path for studying the large-scale spatial–temporal evolution and influencing factors of China's National Archaeological Site Parks. Finally, through a systematic study of the spatial–temporal development and distribution patterns of National Archaeological Site Parks, this work reveals the spatial–temporal evolution patterns of China's National Archaeological Site Parks, identifies their driving factors, and provides a scientific basis and data support for improving the overall spatial planning and management system of China's National Archaeological Site Parks. It also provides a case study on China's National Archaeological Site Parks for the protection and management of large heritage

sites, which can also serve as a useful reference for heritage protection, especially the protection of large heritage sites, in other countries.

This study scientifically analyzes the spatial–temporal evolution and influencing factors of National Archaeological Site Parks. But, unavoidably, it also has some limitations. First of all, the study did not examine micro-level factors, such as the ages, types, and local characteristics of the National Archaeological Site Parks themselves, which are equally important for the planning, construction, and sustainable development of National Archaeological Site Parks. Future research should focus on National Archaeological Site Parks in a specific region, combining micro-level factors with existing research to enhance the study of these spatial and temporal patterns. Secondly, due to limitations in the research methodology and data, this paper does not discuss the interactions and grouping logics between different influencing indicators and between individual National Archaeological Site Parks, which calls for collaboration with governmental departments to carry out more in-depth research in this area. Finally, the research model developed in this study is intended to study the feasibility of the spatial–temporal patterns of National Archaeological Site Parks on a large scale. In the future, this model can be optimized and expanded to a more local scale to construct a comprehensive model suitable for analyzing the spatial–temporal evolution patterns and influencing mechanisms of National Archaeological Site Parks on different scales.

**Supplementary Materials:** The following supporting information can be downloaded from the National Archaeological Site Parks of Yinxu (http://www.anyangyinxu.cn/), Haihun Marquis (http://www.hhhhg.com.cn/), Daming Palace (https://www.dmgpark.com/), and Suitang dynasty Luoyang city (https://www.suitangluoyang.com/) websites: Figure 2: China's National Archaeological Site Parks.

**Author Contributions:** Conceptualization, Y.X., T.L. and S.C.; methodology, T.L. and S.C.; software, T.L. and X.Z.; validation, Y.X. and S.C.; formal analysis, T.L. and X.Z.; investigation, T.L., X.Z. and S.Q.; resources, Y.X. and T.L.; data curation, T.L., X.Z. and S.Q.; writing—original draft preparation, T.L. and X.Z.; writing—review and editing, Y.X., T.L., S.C. and Y.D.; visualization, S.C., T.L., S.Q. and Y.D.; supervision, Y.X. and S.C.; funding acquisition, Y.X. All authors have read and agreed to the published version of the manuscript.

**Funding:** This research was funded by the National Social Science Foundation Project "Research on the Path of Public Perception Improvement of National Archaeological Site Park", grant number (19BKG047). The APC was funded by Yueting Xi.

**Institutional Review Board Statement:** Not applicable.

**Informed Consent Statement:** Not applicable.

**Data Availability Statement:** The raw data supporting the conclusions of this article will be made available by the authors upon request.

**Acknowledgments:** Many thanks are extended to Xiaobo Zhang, Sha Luo, Yuru Han, and Wenbo Ma for their help.

**Conflicts of Interest:** The authors declare no conflicts of interest.

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
