# Peer review of "A Study of the Spatial–Temporal Development Patterns and Influencing Factors of China’s National Archaeological Site Parks"

_sustainability, doi:10.3390/su16083397_

Round 1
Reviewer 1 Report
Comments and Suggestions for Authors
1. It is essential to revise the abstract to be more concise and focused. The revised abstract should emphasize the rationale behind conducting the study and highlight its significance.
2. Section 2.1 should be condensed and integrated into Section 1, while Section 2.2 is not essential.
3. The name of section 3 should be titled “Materials and methods”.
4. The authors should provide a detailed explanation for each indicator listed in Table 1.
5. As mentioned in section 4.1.1, age serves as a crucial criterion for selecting national archaeological site parks. Are there official criteria established for the selection of national archaeological site parks? If such criteria exist, they should all be considered as influencing factors.
6. In Section 4.2, depicted in Figure 7, it is evident that a higher number of national archaeological site parks are located in areas with long-standing human habitation. Therefore, the historical factor is also deemed significant and warrants discussion.
Comments on the Quality of English Language1. The language of this manuscript has many grammatical errors, substantial polish must be made.
Reviewer 2 Report
Comments and Suggestions for Authors
In the paper the development of Archaeological site Parks take into account the influence of heritage tourism as driving force and other factors but no mention is made about the air pollution increase due to uncontrolled urban development as well as to industrialization occurring nearby cultural heritage sites. These phenomena represent a risk for the preservation of archaeological sites. Another problem is the sharp increase of heritage tourism which must be counterbalanced by a sustainable policy of preservation of archaeological remains. Not enough emphasis is given to the priority of the preservation of cultural relics under the pressure of the sharp increase of heritage tourism and the air pollution ascribed to urbanization of large cities nearby heritage sites.
Reviewer 3 Report
Comments and Suggestions for Authors
The article topic is ‘A Study of Spatial-Temporal Development Pattern and Influencing Factors of China’s National Archaeological Site Parks’. There is a lack of justification for the research question and novelty. In addition, the article has serious flaws, additional experiments needed, research not conducted correctly. Further, the article is weak in terms of research aim, research methodology, data sampling, research design, paper structure, readability, and novelty.
1)Abstract: unclear what the paper is about and how to conduct, i.e., lack of research aim and method including sampling for data collection and approach and techniques for data analysis.
2)Lack of a clear research aim throughout the manuscript.
3) Lack of a 'Method' Section that indicate that the article lacks of a sound research methodology, although Section 3. ‘Research Strategies’ actually is a section partly describing the technical data collection process.
4) Lack of a clear justified research question that is for what and how to associate with research aim that is key to the topic. As such, the topic has not been conducted fully.
5) As such, Section 4 'Results' section lacks clear association with ‘Method’ Section that is missing currently..
6) In 'Discussion' section, Lack of clear discussions that are based on the results of Section 4 ‘Results’ to integrate and reveal interesting schemes, which also need to discuss clear contributions of the results of the paper, and research limitations that are currently missing.
7) Hence, Section ‘6.Conclusions’ is unjustified.
Extensive editing of English language required
Round 2
Reviewer 1 Report
Comments and Suggestions for Authors
The revised manuscript addressed all my concerns, I think this manuscript could be acctepted now.
Reviewer 3 Report
Comments and Suggestions for Authors
The paper has been improved.
Comments on the Quality of English LanguageMinor editing of English language required
